# Interactive Text Generation

**Felix Faltings**[1*]   **Michel Galley**[2]   **Kianté Brantley**[3]   **Baolin Peng**[2]
**Weixin Cai**[2]   **Yizhe Zhang**[2]   **Jianfeng Gao**[2]   **Bill Dolan**[2]
[1]MIT  [2]Microsoft  [3]Cornell University

## Abstract

Users interact with text, image, code, or other editors on a daily basis. However, machine learning models are rarely trained in the settings that reflect the interactivity between users and their editor. This is understandable as training AI models with real users is not only slow and costly, but what these models learn may be specific to user interface design choices. Unfortunately, this means most of the research on text, code, and image generation has focused on non-interactive settings, whereby the model is expected to get everything right without accounting for any input from a user who may be willing to help. We introduce a new *Interactive Text Generation* task that allows training generation models interactively without the costs of involving real users, by using user simulators that provide edits that guide the model towards a given target text. We train our interactive models using Imitation Learning, and our experiments against competitive non-interactive generation models show that models trained interactively are superior to their non-interactive counterparts, even when all models are given the same budget of user inputs or edits.

## 1   Introduction

Advances in generative modeling have made it possible to automatically generate high-quality texts (Brown et al., 2020), code (Chen et al., 2021), and images (Ramesh et al., 2021), but these creations can be unsatisfactory in many respects. For example, they often suffer from content errors—e.g., hallucinations (Ji et al., 2022)—that may require help from the user. Even if the generation is of good quality, it may not be the kind of text, code, or image that the user was hoping to obtain. Indeed, open-ended and complex generation tasks are often underspecified (e.g., from a simple prompt), which makes it almost impossible for the model to

---
*Corresponding authors: faltings@mit.edu, {mgalley, jfgao}@microsoft.com. Yizhe Zhang is currently at Apple.

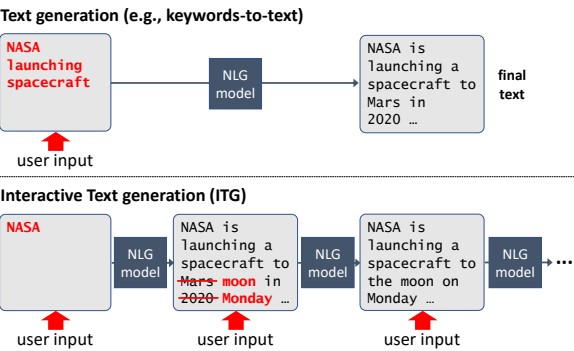

Figure 1: The ITG framework of this paper allows generation models to be trained and evaluated by directly interacting with users or user simulators.

satisfy user needs without additional information. Distinguishing the real user need from the need as initially presented to the system has been the focus of decades of research (Taylor, 1968), and usually requires *interacting* with users to clarify their needs (Gao et al., 2022). Unfortunately, much of the research in generative models have been in "one-shot" settings, which don't allow any kind of iterative refinements to steer the generation towards what the user really wants.

This paper introduces a new end-to-end generation task, *Interactive Text Generation*, that accounts for interactivity without requiring real users during training. The framework is illustrated in Figure 1, where the model and a "user" take turns editing the text until a given stopping criterion is met. As this setup would be impractical to train with real users, we rely on a user simulator that provides a few high-quality edits that guide the model towards a given target text. This interspersion of user edits in the generation process allows text generation models to more efficiently use inferential capabilities of large language models (LLM). Contrast interactive text generation in Figure 1 with conventional text generation, where both systems are given exactly three user input words. Interactive text generation leverages LLM's capability to infer, e.g., "space-

craft" from "NASA" and allows it to focus on parts of the text (e.g., "Monday") that are more difficult to predict, and this allows the interactive approach to generate text that better meets user's needs.

Our work makes the following contributions:

- We propose a task and framework for interactive text generation, and release models, a dataset, and user simulators. Crucially, this task evaluates generation models with the same budget of user inputs or edits, to ensure the comparison between interactive and non-interactive models is fair.
- We present methods to train Transformer-based (Vaswani et al., 2017) interactive text editing models using imitation learning, where our models learn to imitate an expert that dynamically constructs a trajectory from the current document state to the goal state (target document). Our editing models include both autoregressive and non-autoregressive versions, with the non-autoregressive one achieving the best results.
- We show that interactivity indeed does help thanks to imitation learning when compared to their non-interactive counterparts, across different evaluation metrics, model types, and model sizes and with the same amount of user inputs. This finding is consistent with prior work showing that user needs are often best handled interactively (Oddy, 1977; Belkin et al., 1995; Radlinski and Craswell, 2017; Gao et al., 2022), and confirms that our benchmark helps quantify the benefit of interactivity in text generation.
- As user simulation is a key component of this new task, we contribute different user simulators. Our experiments show performance of our models remains consistent with different user simulators, which highlights the robustness of our new benchmark.

Another contribution is that text generation in this work is not merely aimed at producing well-formed text, but also at creating text that is tied to user needs. We release this framework with the hope it will help researchers in NLP, Imitation Learning, Reinforcement Learning, and AI in general as it provides an environment to train AI agents that directly interact with users and user simulators.[1]

## 2 Task: Interactive Text Generation

The task introduced in this paper considers a simple setting in which the system and user collaboratively write a document. As our goal is to train generation models that can do most of the heavy lifting, this task gives a more directional role to the user, while the bulk of the text is generated by the system. Therefore, our task is similar to the instructor-executor frameworks (Hu et al., 2019; Kiseleva et al., 2022) seen in other tasks. In the case of text generation, motivational examples of such instructor-executor interactions include a student writing a paper while getting occasional feedback from an advisor, or a freelance writer getting instructions from a client.

Our task models the interaction between a user and system, where the two parties successively take turns making changes to a draft document. As this interaction applies to both training and testing, it would be unrealistic to assume we have real users in both cases, and we therefore rely on user simulators. Although building effective user simulators is notoriously hard in tasks such as dialog (Schatzmann et al., 2007; Li et al., 2016; Lin et al., 2022; Gao et al., 2019), it is less so in our work given how we frame the interactive generation task: We assume that there is a goal text which one can think of as representing the user's desire or needs. The system does not know the goal, but the user simulator does. The objective of the agent is to get as close to the goal text as possible, while minimizing the number of edits the user simulator needs to make.

This framing makes it much easier to design effective user simulators, as a comparison between the current draft and goal text can help infer useful text edits. While agents in this setup are given an advantage by interacting with an oracle that has knowledge of the goal document, the number of oracle edits is generally set to be small, and we ensure comparisons between all models (including non-interactive models) are fair by giving each model the same budget of edits or inputs derived from the goal document.

### 2.1 Task Formulation

We can formalize our task by the following protocol. There are two players in the game, the agent and user. Starting from a blank document, the players take turns producing drafts over $H$ rounds.[2] The user has a goal document $G$, and the objective of the game is to get as close to the goal as possible. Closeness is quantified by a scoring function

---

[1]Code and models for this paper will be made public.

[2]Except for index ranges, e.g. $t = 1, ..., T$, we will use capital letters to denote random variables, and lower case letters to denote realizations.

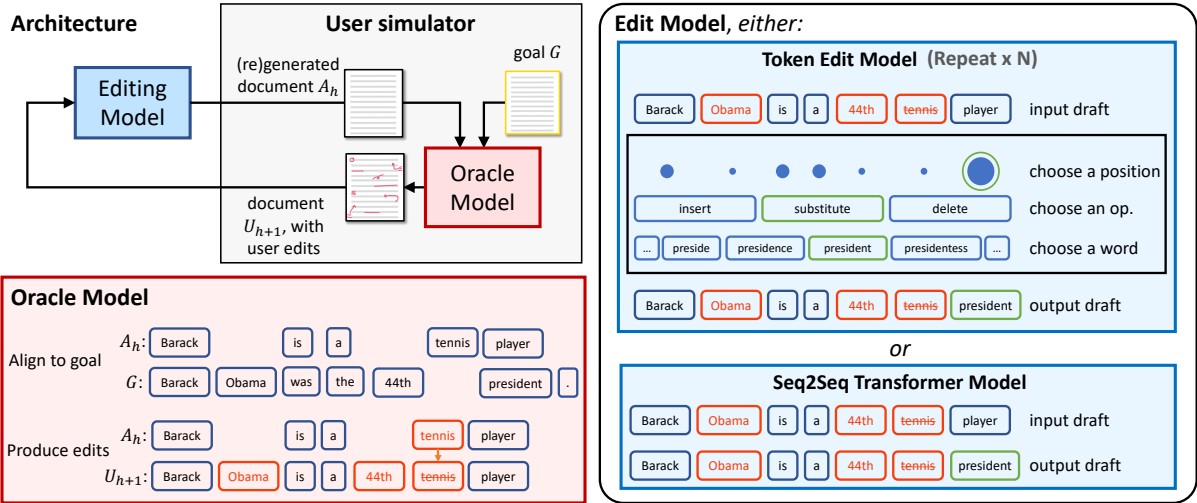

Figure 2: Model architecture: the current draft repeatedly goes through two phases. First, it is edited with the edit model (either a non-autoregressive token edit model or a standard seq2seq Transformer), and then it is annotated with edits coming from the user simulator. While the user simulator has access to the gold document, all our generation systems (interactive and non-interactive) are evaluated with the same budget of user inputs.

$s(\cdot; G)$, and a tolerance $\delta > 0$. We denote the agent draft at step $h$ by $A_h$, and the user draft at step $h$ by $U_h$. If we then fix a goal document $G$, the protocol is as follows. For $h = 1, ..., H$:

1. User observes $A_{h-1}$. If $s(A_{h-1}; G) > 1 - \delta$, stop. Otherwise, the user produces $U_h$ according to $G$ and $A_{h-1}$.

2. Agent observes $U_h$ and produces $A_h$.

Here $A_0$ is a blank document. The tolerance $\delta$ represents how close the agent needs to get to the target for the user to be satisfied. Let $T \leq H$ be the time at which the interaction stopped. We evaluate the task by looking at $s(A_T; G)$ and $T$. The higher $s(A_T; G)$, the better the produced document, and the lower $T$, the more time the user saved.

We assumed here that each instance of the task is parameterized by a goal document $G$. However, one could instead use multiple goal documents, $G_1, ..., G_M$, or even replace $s(\cdot; G)$ with some general score function $s(\cdot)$ that is independent of any goal document, which one could think of as a utility function for the user.

In this work, we assume that the user behavior is fixed and we will learn an agent policy.

## 3 User Simulator

This section describes the user simulator used in our work. Throughout this section and the next, we refer the reader to the Appendix for more details.

Where appropriate, we indicate references to the appendix in parentheses.

### 3.1 Edits

Both our simulated user described below, and our model described in Section 4, operate by making edits to a document. We will consider single-word edits of three types: inserting a word, deleting a word, or substituting one word for another. If we denote an edit by $e$, and a document by $x$, then $y = [e](x)$ will denote the action of $e$ on document $x$, producing a new document $y$. Note that $y$ will differ from $x$ by a single word. If we have multiple single-word edits $e_1, ..., e_N$, we can apply them one after another, as in $y = [e_1...e_N](x)$.

### 3.2 Simulated User

Because the user has knowledge of the goal document $G$, we can easily define a sensible user simulator. Given a current draft produced by the agent $A_h$, we can compute sequences of edits $e_1, ..., e_N$ such that $G = [e_1...e_N](A_h)$, as detailed in Section 3.3 below. The user will produce a draft $U_{h+1} = [e_1...e_n](A_h)$ only by applying some small number of these edits $n \leq N$, where the edits are chosen according to a heuristic (App. B). An example heuristic would be to pick edits $e_1, ..., e_n$ that insert informative words first. This way, the user provides information to the agent by producing a draft $U_{h+1}$ that is a few steps closer to $G$ than $A_h$. This is like providing a gradient in the direction of $G$. Note that while this represents one

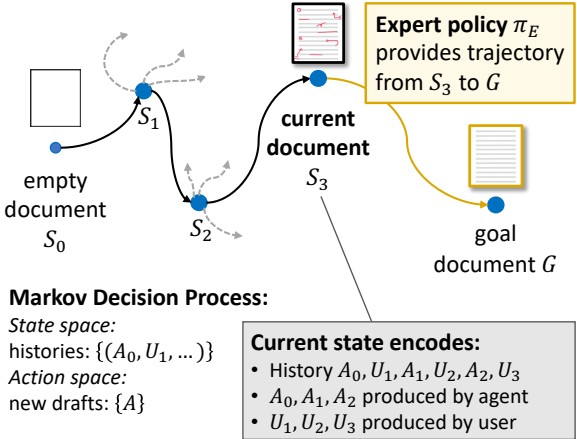

Figure 3: Interactive text generation as imitation learning: The editing model (agent policy $\pi_\theta$) is trained to imitate the expert policy $\pi_E$.

way to build a user simulator, our task formulation is completely agnostic to the user behavior.

In our experiments we ensure that comparisons between non-interactive and interactive systems are fair by fixing the total number of single-word edits made by the user at $N$. In the non-interactive case, all $N$ edits are made at the start. In the interactive setting, the $N$ edits are spread out over $M$ rounds of editing, where at each round the user only makes $N/M$ edits.

### 3.3 Alignments

For any $x, y$ there may be many sequences $e_1, ..., e_N$ such that $y = [e_1...e_N](x)$. There is always such a sequence since we can delete $x$ completely and generate $y$. Practically, we can compute a *set* of such sequences of edits $e_1, ..., e_N$ by finding an alignment between $x$ and $y$, as illustrated in Figure 2. The alignment defines which words in $x$ map to words in $y$, and which words were deleted or inserted. Words that match to identical words are conserved, whereas mismatches correspond to substitutions. From there it is easy to get the edits $e_1, ..., e_N$ that transform $x$ into $y$. We will denote this set by $A(x, y)$. We use contextual word embeddings from BERT to compute these alignments, so that words in $x$ map to words in $y$ with similar meaning, as we found that this tends to produce more natural alignments (App. A.3).

## 4 Imitation Learning Model

The agent policy $\pi_\theta$ will parameterize a distribution over documents, so that at each step of the interaction we draw $A_h \sim \pi_\theta(\cdot|S_h)$. Here $S_h$ is the his-

tory of drafts $(A_0, U_1), \ldots, (A_{h-1}, U_h)$ exchanged so far. This information is necessary for the agent to make inferences about the goal $G$. However, this history can become very long, especially since each element is an entire document. Therefore, in practice we only give the policy $\pi_\theta$ access to the last set of drafts, $(A_{h-1}, U_h)$, which we represent as a diff (App. B), as illustrated in Figure 2.

### 4.1 Left to Right Autoregressive Model

The first agent model that we consider is a simple sequence to sequence (S2S) model that generates outputs in a left-to-right autoregressive manner,

$$\pi_\theta(A_h = a|S_h) = \prod_{i=1}^{H} \pi_\theta(a_i|a_{<i}, S_h),$$

where $a = (a_1, ..., a_L)$ and $a_{<i} = (a_1, ..., a_{i-1})$. Note that this model autoregressively generates the new draft from scratch.

### 4.2 Token Editing Model

While our S2S model forms a reasonable baseline, we note that it is not particularly adapted for the task. Instead, we also propose a token editing model that directly edits the previous draft $U_h$ to produce its revised draft $A_h$ by making a series of edits. This way, both the agent and user operate by making edits. Additionally, we add a stopping action to allow the model to decide when to stop editing. Concretely, we parameterize the model as a distribution over edits (and the stopping action), $\pi_\theta(\cdot|[e_1...e_{t-1}](U_h), S_h)$. In other words, based on the previous edits it made, and the current state, the model decides what edit to make next. The probability of producing a particular draft $a$ is,

$$\sum_\tau \pi_\theta(\text{stop}|a, S_h) \prod_{k=1}^{M} \pi_\theta(e_k|[e_1...e_{k-1}(U_h), S_h),$$

where the sum is over all sequences of edits $\tau = (e_1, ..., e_M)$ such that $[e_1...e_M](U_h) = a$. See Figure 2 for an illustration.

### 4.3 Training

We train our model to copy an expert policy $\pi_\theta$ that would perform well in our task. Because this expert is only used at training time, it is very simple: it produces the goal $G$. See Figure 3 for an illustration. We follow the DAgger (Ross et al., 2011) algorithm as outlined in Algorithm 1, which tries to minimize the following objective:

$$\mathcal{L}(\theta) = \mathbb{E}_{G\sim\nu}\mathbb{E}_{S_h\sim\pi}\left[-\log\pi_\theta(G|S_h)\right],$$

**Algorithm 1:** DAgger

Initialize $\pi_\theta$;
Initialize $\mathcal{D} \leftarrow \emptyset$;
$\beta \leftarrow 1$;
**repeat**
  $\pi \leftarrow \beta\pi_E + (1-\beta)\pi_\theta$;
  Sample goal documents $g_1, ..., g_C \sim \nu$;
  Sample trajectories from $\pi$ with
    contexts $g_1, ..., g_C$;
  Collect $B$ states from sampled
    trajectories into $D$;
  Aggregate $\mathcal{D} \leftarrow \mathcal{D} \cup D$;
  Train policy $\pi_\theta$ on $\mathcal{D}$;
  $\beta \leftarrow \lambda\beta$;
**until** *convergence*;

where $\pi$ is the roll-in, or sampling policy, $\nu$ is a distribution over goals, and the second expectation is over histories produced by running our task protocol with the policy $\pi$ as the agent (App. A.4). Setting $\pi$ to $\pi_\theta$ would allow us to train on-policy, but since $\pi_\theta$ will start off as a poor policy that visits many unnecessary states, DAgger uses a mixture between the agent and expert policies, so $\pi = \beta\pi^* + (1-\beta)\pi_\theta$, where $\beta \in [0,1]$ is annealed to 0 during training.

### 4.4 Likelihood Estimation

The training objective from Section 4.3 requires the computation of the negative log-likelihood $-\log \pi_\theta(G|S_h)$. For the autoregressive model, this can be directly computed as a simple product as described in Section 4.1. On the other hand, the likelihood for the token editing model involves a sum over sequences of edits which quickly becomes intractable. Instead, we optimize an upper bound on it. Using the alignments defined previously, we can specify a set of edit trajectories, denoted $A(U_h, G)$, that are indexed by permutations $\sigma \in \mathcal{S}_M$, where $M$ is the length of the edit trajectories (App. A.3). So, an edit trajectory in $A(U_h, G)$ will be written $e_1^\sigma, ..., e_M^\sigma$, and $[e_1^\sigma, ..., e_M^\sigma](U_h) = G$ for all $\sigma$. We can then assume that most alternative trajectories[3] will have low probability, so we restrict the sum to all trajectories in $A(U_h, G)$. For convenience, we will write $X_k^\sigma = [e_1^\sigma, ..., e_k^\sigma](U_h)$. This gives the

---

[3] For example the infinite number of trajectories that involve inserting and deleting the same word.

upper bound (App. A.5),

$$-\mathbb{E}_k\mathbb{E}_{\sigma_{1:k}}\left[\frac{M+1}{n_k}\sum_{\sigma_{k+1}}\log\pi_\theta(e_{k+1}^\sigma|X_k^\sigma, S_h)\right],$$

where the two expectations are over uniform distributions, and $n_k$ is the number of terms in the sum. The first expectation ranges over prefixes of edit sequences $e_1^\sigma, ..., e_k^\sigma$ for $\sigma \in \mathcal{S}_M$, and the sum ranges of over distinct edits $e_{k+1}^\sigma$. In words, this objective says that we select a random number $k$ of edits, in random order $\sigma$, that move us toward $G$, giving us the intermediate draft $X_k^\sigma$. We then try to predict the next edit that brings us closer to $G$. We evaluate this objective stochastically as described in Algorithm 2 (App. A.5).

### 4.5 Decoding

When decoding from the autoregressive model, we use standard algorithms such as beam search. For the token editing model, we sequentially sample edits from the model until the model's probability of stopping reaches a given threshold or we exceed a maximum number of decoding steps. If we hit the timeout, we return the draft that had the highest stopping probability according to the model. This ensures that whatever draft the agent returns it is highly confident that it is a finished document. For example, if the model were editing "the man" into "the man and the dog", we would not want to stop and return the draft "the man the dog" where the model hasn't yet added the word "and". Algorithm 4 outlines this procedure (App. A.7).

## 5 Experiments

Because of the novelty of our task, the main goal of our experiments is to assess the benefit of interactivity in text generation. We also provide example interactions between the learned agent and user simulator. Ablations and additional details on experiment settings are in the appendix.

### 5.1 Setup

**Data** We consider single sentences from CNN/DailyMail article summaries (Nallapati et al., 2016). While we only consider single sentences for ease of implementation, we can extend our models to full documents in a straightforward way. Using news articles gives complex and factually grounded sentences, while restricting our attention to the summaries keeps the sentences

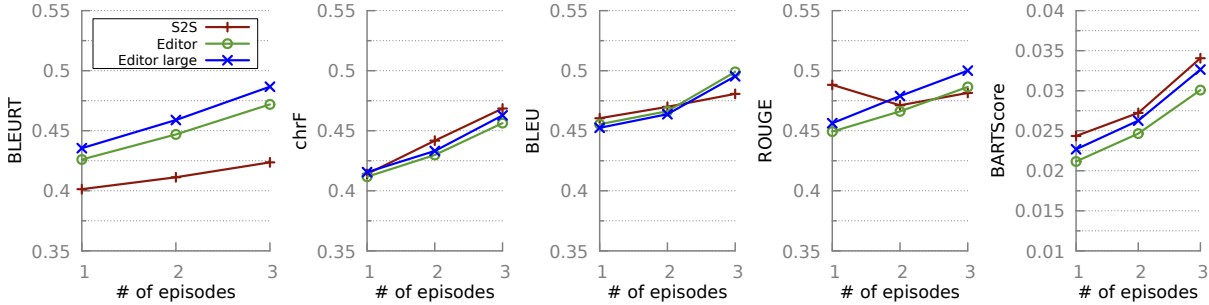

Figure 4: Results showing the benefit of interactive generation, where #episodes=1 means the entire budget of 6 oracle edits is given to the model all at once (i.e., no interactivity). For #episodes=2, the model receives as input 3 oracle edits per episode (2x3=6). For #episodes=3, the model receives 2 oracle edits per episode (3x2=6).

self-contained. This dataset forms the distribution over goal documents $G$.

**Model** We implement our models using pretrained transformer models, with additional MLP heads to predict the edit actions in the case of the token editing model.

**Metrics** We evaluated generation using BLEU (Papineni et al., 2002), CHRF (Popović, 2015) BLEURT (Sellam et al., 2020), BARTSCORE (Yuan et al., 2021), and ROUGE (Lin, 2004). As BARTSCORE returns a score interpretable as a log-probability, we report the natural exponent of that score. In the case of both BLEU and ROUGE, we evaluate with their unigram versions (BLEU-1 and ROUGE-1) as they provide interpretability of the results at the token level.

### 5.2 Interactivity

To probe whether interactivity helps, we compared the performance of our model given different levels of interactivity. Concretely, we take a fixed number of edits $N$ provided by the user, and compare the performance of the model when those edits are provided over a varying number of episodes $M$. Thus, for given $M$, the user and agent interact over $M$ episodes, with the user making $N/M$ edits at each episode. Note that the total amount of information provided by the user in each setting is thus the same. The only difference is that in interactive settings the agent is able to make changes in between receiving edits from the user. While this setup is not able to probe the advantages that interactivity provides in terms of human-computer interaction, we still expect to see better performance in the interactive case. For example, the model may be able to preempt some of the edits the user makes, allowing the user to use its budget to make other edits.

## 6 Results

**Automatic evaluation** Figure 4 presents our main results on interactivity. We can see that for our main model, splitting the same number of user edits over more episodes leads to better scores across BLEU, BLEURT and BERTSCORE. For example, comparing the setting where the model receives all 6 user edits at the start in one episode, against the setting where the edits are given across 3 episodes, we see improvements of about 7%, 4% and 5% (absolute % gains) in terms of BLEU, BLEURT, and BERTSCORE respectively. While these differences are not large in absolute terms, we emphasize that this gain comes exclusively from interactivity. The amount of information given to the model is the same in both settings. This suggests that, even in this narrow sense, interactivity helps. Note that there may also be many more benefits from a human-computer interaction standpoint that we cannot quantify here.

We also note that our token editing model (Editor) outperforms the left-to-right, autoregressive sequence to sequence (S2S) model[4]. While the difference is not staggering, it is notable given the success of standard S2S models across a variety of text generation tasks. As motivated in Section 4, the token editing model is more naturally suited to the task, and we argue that it constitutes the starting point for more interesting interactive models. For example, one could foresee training the model using reinforcement learning, in which case

---

[4]In the case of the S2S model, we added a word reward feature (He et al., 2016), which is sometimes called "length penalty". We tuned this feature on a validation set of 500 instances. We added this feature due to the observation that S2S outputs in our task are often too short, which has been noted in other tasks such as machine translation (Murray and Chiang, 2018). This feature was only necessary with the S2S model, and resulted in the three models of Figure 4 having average lengths close to one another (within 6%).

|  | Non-interactive | | = | Interactive | |
|---|---|---|---|---|---|
|  | ++ | + | | + | ++ |
| Meaning | 6.7% | 30.2% | 9.6% | **42.6%** | **10.9%** |
| Fluency | **8.0%** | **36.0%** | 15.1% | 33.4% | 7.5% |

Table 1: Human evaluation using Editor Large model, comparing interactive generation (3 episodes, 2 oracle edits) vs. non-interactive (1 episode, 6 oracle edits). Results indicate judges' preference: ++ (definitely superior), + (somewhat superior), = (equal quality).

the structure of the editing model based on editing *actions* is better suited than a S2S model.

**Human evaluation** We conducted a human evaluation (Table 1) to compare interactive and non-interactive generation, similar to the automatic evaluation in Figure 4. We selected the best-performing model (BART Large) and compared 3-episode generation against 1 episode on 1k test instances. Each instance was evaluated by 5 mechanical turkers. The judges were asked to compare each pair of generations based on semantic similarity to the gold text ("meaning") and linguistic quality ("fluency") using a 5-point Likert scale. Table 1 shows that interactivity improves performance, as the interactive system is generally semantically closer to the target (significance at $p < 1e - 7$). The interactive system exhibits slightly lower fluency on average, although the level of significance is weaker here ($p = .037$). We hypothesize this slight decrease in fluency is due to multiple rounds of generation.

**Ablations** We provide extensive ablations of model variants. As a benchmark for comparison, we look at the quality of the text produced by our models after interacting with the user over several episodes. For better comparisons we use a fixed number of episodes and a fixed number of user edits per episode. We use 3 edits and 4 episodes. Tables 2 and 3 present our ablations. Note that these results use 4 episodes, with 3 user hints per episode (a total of 12 user hints) compared to the 6 total user hints in Figure 4, so the overall results are higher.

The baseline model is trained with a noise level of $\sigma = 0.3$, an unrestricted user and a sampling annealing rate of 0.9. All models in Table 2 were evaluated with the adjacent and contiguous user heuristics. Table 2 presents variations on training parameters. The noise level is the amount of noise injected during training, the user is the (sole) heuristic used for the user during training, and the

| Model | BLEU | BLEURT | BART |
|---|---|---|---|
| Bart Editor (baseline) | 0.76 | 0.70 | 0.14 |
| *noise level* | | | |
| 0.0 | 0.74 | 0.69 | 0.14 |
| 0.1 | 0.78 | 0.73 | 0.16 |
| 0.2 | 0.73 | 0.67 | 0.12 |
| *oracle* | | | |
| contiguous | 0.71 | 0.66 | 0.12 |
| adjacent | 0.60 | 0.61 | 0.09 |
| adj+config | 0.66 | 0.64 | 0.11 |
| *Sampling annealing rate* | | | |
| 0.85 | 0.76 | 0.70 | 0.14 |
| 0.80 | 0.71 | 0.67 | 0.12 |

Table 2: Ablation results with different *training* hyperparameters. All ablations results are relative to the baseline. BART stands for BARTSCORE.

| Model | BLEU | BLEURT | BART |
|---|---|---|---|
| adj+config (baseline) | 0.76 | 0.70 | 0.14 |
| contiguous | 0.76 | 0.69 | 0.14 |
| adjacent | 0.78 | 0.75 | 0.19 |
| unrestricted | 0.78 | 0.75 | 0.18 |

Table 3: Ablation results with different oracles changed at *test* time.

sampling annealing rate indicates how quickly we anneal from the expert to the trained model while sampling (lower is faster). Table 3 compares different user heuristics at *test* time.

We note that adding noise during training improves results (e.g. noise level 0.1 vs. 0.0), while annealing too fast can hurt performance (annealing rate 0.8 vs. baseline). Interestingly, training with a user that better matches the inference-time user leads to worse performance (e.g. adj+contig vs. baseline). It seems that using the most informative user (which simply returns the most informative edits, without restriction) leads to the best model (baseline). Comparing different user simulators at inference time, we see that adding restrictions to the user leads to decreased scores, as expected. Interestingly, we see that the most impactful restriction seems to be enforcing contiguous edits. We suspect that this is because contiguous edits are highly predictable. For example, predicting "Obama" after "Barack" is fairly obvious. Thus, if the user didn't provide contiguous edits, and only inserted the word "Barack", the model could score an easy win by predicting "Obama".

**Examples** Table 4 provides two examples of interactions between the agent and user. We emphasize that these examples were *not* cherrypicked. Note how the agent is able to revise its previous version of the text based on the new information provided by the user at each step. Qualitatively, these interactions could be much more natural for a user than the one shot setting that is prevalent in the literature. However, a systematic evaluation of this claim requires a more comprehensive user study that lies out of the scope of this work.

# 7 Related Work

**Text Generation** Prior work on natural language generation (NLG) has largely focused on non-interactive settings that have become increasingly more challenging and open-ended, e.g., with generation from prompts (Fan et al., 2019), outlines (Rashkin et al., 2020), topics or keywords (Ghazvininejad et al., 2016; Yan, 2016), plots (Riedl and Young, 2010), descriptions (Jain et al., 2017), events (Martin et al., 2017). This increase of difficulty can make NLG more prone to content quality issues, such as hallucinations (Wiseman et al., 2017; Filippova, 2020; Çelikyilmaz et al., 2020; Ji et al., 2022), that can require post-editing from the user. Several works explored ways for LLMs to improve their outputs by iteration and self-critiquing (Huang et al., 2022; Gou et al., 2023; Welleck et al., 2022). In particular, Welleck et al. (2022) presented models for text generation and self-correction that also incorporated external feedback. However, in their case the feedback is used at training time to learn a corrector. In our task, the user feedback comes at inference time and the agent must use that feedback to guess what the user would like to generate.

**Non Autoregressive Generation** Several works considered non-autoregressive text generation (Gu et al., 2019; Shen et al., 2020; Xu and Carpuat, 2021; Stern et al., 2019; Zhang et al., 2020; Welleck et al., 2019), but these models all focus on one-shot text generation. While some models are able to edit text (Gu et al., 2019; Xu and Carpuat, 2021), it is primarily used as a means to refine the model's generations. On the other hand, we consider editing text into a completely different version conditioned on a given set of user-provided edits.

**Text Editing** Text editing has previously been considered from two different angles. On the one

hand, various works (Zhang et al., 2019; Du et al., 2022b) have studied the types of revisions made by humans. On the other hand, works have focused on modeling text edits (Guu et al., 2018; Yin et al., 2018; Faltings et al., 2021; Akoury et al., 2020), but they have generally been restricted to modeling a single episode of user edits at a time. In our framework, model edits and user edits are interleaved. We note that (Du et al., 2022a) presented an interactive revision system, but their model was nevertheless trained on a corpus of edits, rather than in an interactive environment as in our case. The recent versions of GPT-3 (Brown et al., 2020; Ouyang et al., 2022) and ChatGPT[5] also offer text editing capabilities. For example, ChatGPT can handle a prompt containing a piece of text and instruction to improve the text, and ChatGPT's ability to execute that command is often quite impressive. The ability of ChatGPT to interact with users was somewhat explored in (Bang et al., 2023), although not in the context of text editing. We think our work is complementary to GPT-3 and ChatGPT, as we provide a framework for both modeling and evaluating edit models in a more end-to-end setting. Our model of interaction between user and system, where the user and system are both editing the text, may also be more natural than dialogue. Ultimately, we think it will be beneficial to fine-tune very large language models such as GPT-3 in an environment that exposes them to interaction with a user or a user simulator (i.e., akin to user-in-the-loop training). This benefit is currently somewhat captured using reinforcement learning (RL) to tune large language models from human feedback (Ouyang et al., 2022), except that our approach features an actual environment representative of the end-user experience while Ouyang et al. (2022) is more akin to offline RL.

**Interactivity** Several works have explored interactivity between humans and models to complete tasks collaboratively. Lee et al. (2022) presented a dataset to reveal large language models' capabilities in assisting creative writing. Zhang et al. (2022); Li et al. (2022) explored pre-trained language models for code editing based on human-written comments. Closer to our work, Lahiri et al. (2022) created an interactive framework to refine user intents through test case generations and user feedback, and Kiseleva et al. (2022) studied an interactive agent that can interact with humans

---

[5]https://chat.openai.com/chat

| Example 1 | Example 2 |
|---|---|
| target: Progress 59 spacecraft will re-enter Earth's atmosphere in a week, Russia space agency says . | target: The Large Hadron Collider (LHC) begins again after a two-year shutdown . |

*oracle*: Progress 59 re-enter

agent: Progress in Europe means he has a 59 per cent chance to re-enter the EU .

*oracle*: Progress in Europe means he has a 59 per ~~cent~~ spacecraft will re-enter the EU .

agent: Progress ~~in~~ will help the ~~Europe~~ ~~means~~ ~~he~~ ~~has~~ ~~a~~ 59 million ~~per~~ ~~cent~~ spacecraft will re-enter ~~the~~ Earth ~~EU~~ .

*oracle*: Progress will help the 59 ~~million~~ spacecraft will re-enter Earth's says .

agent: Progress comes after Boeing ~~will~~ ~~help~~ ~~the~~ 59 spacecraft will re-enter NASA ~~Earth~~'s spacecraft, it says .

*oracle*: Progress comes after Boeing 59 spacecraft will re-enter NASA's spacecraft, Russia space agency says .

agent: Progress comes as Boeing's ~~after~~ ~~Boeing~~ 59 spacecraft will re-enter Earth ~~NASA~~ ~~spacecraft~~, Russia's space agency says .

---

*oracle*: (LHC)

agent: Curators at the Mars Space Telescope (LHC) will launch in space the next month .

*oracle*: The Large ~~at~~ the Mars Space Telescope (LHC) will launch in space the next month .

agent: The Large X-ray Telescope ~~at~~ ~~the~~ ~~Mars~~ ~~Space~~ ~~Telescope~~ (LHC) is located ~~will~~ ~~launch~~ in ~~space~~ the Atlantic . ~~next~~ ~~month~~ ~~.~~

*oracle*: The Large X-ray Telescope (LHC) is located in two-year shutdown .

agent: The Large Mars Telescope~~-ray~~ ~~Telescope~~ (LHC) will end a two-year shutdown .

*oracle*: The Large Hadron Collider (LHC) will end a two-year shutdown .

agent: The Large Hadron Collider (LHC) will end a ~~end~~ ~~a~~ two-year shutdown .

Table 4: Example interactions between the oracle and agent. The oracle changes are shown red. Deleted words are crossed out. Note that these are randomly selected examples (i.e., without cherrypicking), with the only curation being the rejection of potentially offensive examples.

and follow natural language instructions to achieve goals in Minecraft. Finally, interactivity has also been studied from a Human-Computer Interaction viewpoint (Clark et al., 2018; Shen and Wu, 2023).

# 8 Conclusions

We presented a new task and benchmark for text generation that operationalizes *interactivity* between an agent and a user, without the need to involve real users during training. Our framework compares interactive and non-interactive systems that are given the same amount of user inputs, and shows that interactive text generation leads to text of higher quality according to multiple automated evaluation metrics and human evaluations. We also present a non-autoregressive editing model that outperforms a standard sequence-to-sequence Transformer in various settings. All baseline, data, and models of this paper will be made public.

# Limitations

A long-term goal of this research is to enable interactive editing of full documents. For practical reasons and to facilitate adoption, we limited text length to 64 tokens, but we plan to extend our benchmark and released datasets to also include paragraph-level and multi-paragraph texts. Another limitation of our work is that training is done with simulated instead of real users, as training with users in the loop can be extremely slow and costly. To make our approximation of user behavior realistic, our work relies on user inputs that real-world users perform routinely (i.e., word insertion, deletions, and substitutions) even settings that are not assisted with AI. However, we recognize that the behavior of real users in a AI-user collaborative setting may differ from that of our user simulators, and leave the study of such behavior for future work. Finally, while the need for interactivity ap-

plies to any kind of AI creation (e.g., also code and images), we leave the application to other generation tasks for future work. We note, however, that this paper treats text as simple sequences of symbols, and our work could readily be applied to other symbolic creations (e.g., code generation).

## Ethics Statement

Text generation systems, including those relying on large language models, run the risk of generating text that is unsafe (Bender et al., 2021; Bommasani et al., 2021; Weidinger et al., 2021), and this also applies to generation models developed in this work. While we have not observed generations that are overtly toxic or hateful, our models can generate texts that are biased and offensive to some readers. As our work focuses on generation of non-fictional texts (in contrast to prior work on story generation), our models also run the risk of generating text that is factually incorrect. However, the focus of our research is to provide interactive capabilities to generation systems, and to make them more in control of the user. As illustrated in Figure 1, a back-and-forth between system and user can make the generated text more factual, and the same kind of interaction can also help increase its safety. Such sanitization of generated texts in our framework may still expose users with unsafe content, so it is still recommended to use current safeguards against hallucinations (Ji et al., 2022), biases (Dhamala et al., 2021; Weinberg, 2022), and other offensive content (Gehman et al., 2020; Welbl et al., 2021; Kiritchenko et al., 2021; Jones et al., 2022; Xu et al., 2022) before displaying text to real users. In that sense, we think our work is complementary to current NLG research on hallucination and safety.

## Acknowledgments

We thank Regina Barzilay, Faeze Brahman, Chris Brockett, Si-Qing Chen, Javier González Hernández, Gerold Hintz, Qiuyuan Huang, Tommi Jaakkola, Zhang Li, Lars Liden, Kosh Narayanan, Hoifung Poon, Victor Quach, Chris Quirk, Sudha Rao, Chenglong Wang, Bin Yu, Zhou Yu, as well as members of the NLP and Deep Learning groups at Microsoft Research for helpful discussions. This work was partly supported by the Eric and Wendy Schmidt Center at the Broad Institute.

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

# A   Methods Details

This section gives more formal and precise definitions of the notions of document, edits and alignments introduced in the main text. We follow by a more detailed description of the token editing model and the derivation of the log-likelihood lower bound.

## A.1   Documents

We consider documents to be elements of $V^L$, the space of strings of length $L$ formed by words in vocabulary $V$. We assume $V$ contains a blank token _, so that $V^L$ corresponds to all documents of length up to $L$.

## A.2   Edits

In this work we consider three types of single-word edits: insertions, deletions, and substitutions. In particular, we do not consider word movements, i.e. changing the position of a word in a document. This will simplify the alignments we use to define the user simulator behavior. We formally define an edit as a 3-tuple specifying: a location $l \in [L] = 1, ..., L$, an operation $o \in \{\text{ins}, \text{del}, \text{sub}\}$, and a word $w \in V$. An edit $e = (l, o, w) \in [L] \times \{\text{ins}, \text{del}, \text{sub}\} \times V$ can then be *applied* to a document $x \in V^L$, which we denote by $[e](x)$, as defined by the following rules:

1. If $o = \text{ins}$: $[e](x) = x_1...x_{l-1}wx_l...x_{L-1}$

2. If $o = \text{del}$: $[e](x) = x_1...x_{l-1}x_{l+1}...x_{L_-}$

3. If $o = \text{sub}$: $[e](x) = x_1...x_{l-1}wx_{l+1}...x_L$

When performing multiple edits $e_1, ..., e_N$ in sequence, we will write[6] $[e_1...e_N](x) = [e_N](...[e_1](x))$ as a shorthand. Note that the edits are not permutation invariant because of the location parameter $l$. For example, if $x$ is a blank document, and $e_1$ and $e_2$ each correspond to inserting "the" and "dog" respectively at the start of the document, i.e. $e_1 = (1, \text{ins}, \text{the})$ and $e_2 = (1, \text{ins}, \text{dog})$, then $[e_2e_1](x) = $ "the dog" whereas $[e_1e_2](x) = $ "dog the".

---

[6] A more operator-like way to write this would have been $[e_N...e_1]$, but we opt for the other order, where the interpretation is that the brackets [] map the sequence of edits $e_1, ..., e_N$ to an operator $[e_1...e_N]$ which is equivalent to performing the edits in sequence.

### A.3 Alignments

Given two documents $x, y$, an alignment will give us a convenient way to find a sequence of edits $e_1, ..., e_N$ such that $[e_1...e_N](x) = y$. We use this to define the user simulator behavior, as well as for training our token edit model. Loosely speaking, an alignment between $x$ and $y$ is a partial mapping between words in $x$ and words in $y$. More formally, it is an undirected bipartite graph with vertex sets $V_x = \{x_1, ..., x_L\}$ and $V_y = \{y_1, ..., y_L\}$, and edges $E$. The edges then match words in $x$ to words in $y$, where some words in $x$ or $y$ may not be matched. A monotonic alignment is one where the edges in the graph do not cross.

For our purposes it will be convenient to represent alignments in a particular way. Rather than a graph, an alignment between $x$ and $y$ will be a pair $\bar{x}, \bar{y} \in V^{2L}$, where $x$, resp. $y$, is a subsequence of $\bar{x}$, resp $\bar{y}$. Moreover, $\bar{x}$ and $\bar{y}$ only contain blank tokens otherwise. See Figure 2 for an example. Then each pair $(\bar{x}_i, \bar{y}_i)$, $i = 1, ..., 2L$ is interpreted as an operation:

1. insertion if $\bar{x}_i =$ _ and $\bar{y}_i \neq$ _

2. deletion if $\bar{y}_i =$ _ and $\bar{x}_i \neq$ _

3. substitution if _ $\neq \bar{y}_i \neq \bar{x}_i \neq$ _

Pairs of blanks (_, _) are ignored. Note that the the positions where neither $\bar{x}_i$ or $\bar{y}_i$ are blank give a monotonic alignment. The converse is not true because we could construct many pairs $(\bar{x}, \bar{y})$ that correspond to the same alignment. This is because we can rearrange the order of the positions corresponding to insertions and deletions since this won't affect the alignment. We therefore require all insertions to come before deletions so that monotonic alignments between $x$ and $y$ correspond one-to-one to pairs $(\bar{x}, \bar{y})$.

Given $x$ and $y$, there may be many possible alignments. We therefore define a score on alignments and choose an alignment that maximizes the score. Given alignment $(\bar{x}, \bar{y})$, the score is

$$S(\bar{x}, \bar{y}) = \sum_{i=1}^{2L} s(\bar{x}_i, \bar{y}_i | x, y),$$

where $s(\bar{x}_i, \bar{y}_i | x, y)$ scores the pair $(\bar{x}_i, \bar{y}_i)$ given the contexts $x$ and $y$. If both $\bar{x}_i$ and $\bar{y}_i$ are blank, the score is 0. If only one is blank, we assign a baseline score $b$. If neither is blank, then $\bar{x}_i$ will correspond to some $x_k$ in $x$. Similarly $\bar{y}_i$ corresponds to $y_l$ in $y$.

We use the cosine similarity between the BERT embeddings of $x_k$ and $y_l$ as their score. We found that using BERT embeddings gave more natural alignments where words of similar meaning or function are matched together. For example, consider two single word documents "red" and "blue". Then it makes more sense to consider the change from "red" to "blue" a substitution rather than a deletion followed by an insertion. On the other hand, "red" to "car" makes less sense as a substitution. This score can be easily optimized using dynamic programming (see for example Gale et al. (1994); Smith et al. (1981); Needleman and Wunsch (1970) for more details).

Given an alignment $\bar{x}$ and $\bar{y}$, the set of indices $A = \{i : \bar{x}_i \neq \bar{y}_i\}$ naturally give a set of edits. We can split $A = I \cup D \cup S$ into the union of indices corresponding respectively to insertions, deletions, and substitutions. These edits can be performed in any order, so the alignment gives a whole set of edit sequences, one for each permutation of $A$. For any $\sigma \in \mathcal{S}_M$ a permutation of $A$, where $M = |A|$ is the size of $A$, there is a sequence of corresponding edits $e_1^\sigma, ..., e_M^\sigma$. The delicate part about getting these edits is determining their location parameters since these will depend on the order $\sigma$. We will need the following two quantities:

1. $I_i^\sigma = |\{j < i : A_{\sigma(j)} < A_{\sigma(i)}, A_{\sigma(j)} \in I\}|$

2. $D_i^\sigma = |\{j < i : A_{\sigma(j)} < A_{\sigma(i)}, A_{\sigma(j)} \in D\}|$

These correspond to the number of insertions and deletions that come before edit $e_i^\sigma$ and which will affect its location parameter. Let $B_i^\sigma = |\{j < A_{\sigma(i)} : \bar{x}_j =$ _$\}|$ be the number of blanks before position $A_{\sigma(i)}$ in $\bar{x}$. Then, for $i = 1, ..., M$, the location parameter will be $l_i^\sigma = A_{\sigma(i)} - B_i^\sigma - D_i^\sigma + I_i^\sigma$. Note that this is just keeping track of where the edit needs to be made in the document $[e_1^\sigma, ..., e_{i-1}^\sigma](x)$ after applying the first $i - 1$ edits. We then treat the three operations separately:

1. $A_{\sigma(i)} \in I$: $e_i^\sigma = (l_i, \text{ins}, \bar{y}_{A_{\sigma(i)}})$

2. $A_{\sigma(i)} \in D$: $e_i^\sigma = (l_i, \text{del}, \_)$

3. $A_{\sigma(i)} \in S$: $e_i^\sigma = (l_i, \text{sub}, \bar{y}_{A_{\sigma(i)}})$

In summary, for documents $x, y$ we can compute a unique alignment $(\bar{x}, \bar{y})$. We then denote the set of edit sequences $\{e_1^\sigma, ..., e_M^\sigma, \sigma \in \mathcal{S}_M\}$ as defined above by $A(x, y)$.

---
**Algorithm 2:** Token Edit Model Objective

  **Data:** $a, s_h$
  Compute edits $A(u_h, a,)$;
  Uniformly sample permutation $\sigma \in \mathcal{S}_M$;
  Uniformly sample $k \in \{1, 2, ..., M+1\}$;
  Compute $x_{k-1}^\sigma = [e_1^\sigma, ..., e_{k-1}^\sigma](u_h)$;
  **return** $\frac{M+1}{n_k} \sum_{\sigma_k} \log \pi_\theta(e_k^\sigma | x_{k-1}^\sigma, s_h)$;

---

### A.4 Markov Decision Process

As stated in Section 2, we fix the user behavior and learn a policy for the agent. To do so, we model our task as a finite-horizon Contextual Markov Decision Process (Sutton and Barto, 1998; Hallak et al., 2015) $(\mathcal{S}, \mathcal{A}, \mathcal{C}, P, R, H, \rho, \nu)$ where $\mathcal{S}$ is the state space, $\mathcal{A}$ is the action space, $\mathcal{C}$ is the context space, $P : \mathcal{S} \times \mathcal{A} \times \mathcal{C} \mapsto \Delta(\mathcal{S})$ is the transition function, $R : \mathcal{S} \times \mathcal{A} \times \mathcal{C} \mapsto [0, \infty)$ is a state-action-context dependent reward function that models the user's utility, $H$ is the horizon, and $\rho \in \Delta(\mathcal{S})$ and $\nu \in \Delta(\mathcal{C})$ are the initial distribution over the state space and a distribution over contexts. The action and context spaces correspond to $V^L$, the sequences of length $L$ over vocabulary $V$, where the context is the goal document $G$. The state space is a history of drafts produced so far. Because the agent is ignorant of $G$, having access to the history of drafts allows the agent to make inferences about the goal $G$. The reward models a tradeoff between minimizing $T$, the time it takes to get to a satisfactory document, and $s(A_T; G)$, the quality of the produced document. The user is modeled by the transition function $P$. Given previous state (i.e. history of drafts) $S_{h-1}$, the agent's draft $A_{h-1}$, and the target $G$, the environment transitions to state $[S_{h-1}|(A_{h-1}, U_h)]$, where $|$ denotes concatenation, and $U_h$ is the user's draft produced according to $A_{h-1}$ and $G$. This framework allows us to use tools like imitation learning to train a policy $\pi_\theta$ for this MDP, as done in Section 4.3.

### A.5 Token Edit Model Likelihood Lower Bound

Recall that at step $h$, the likelihood under the token editing model model of a document $a$, given the current history $S_h$, and latest draft $U_h$ will be,

$$\pi_\theta(A_h = a | S_h) =$$

$$\sum_\tau \pi_\theta(\text{stop}|a, S_h) \prod_{k=1}^M \pi_\theta(e_k | [e_1 ... e_{k-1}(U_h)], S_h),$$

where the sum is over all sequences of edits $e_1, ..., e_M$ such that $[e_1 ... e_M](U_h) = a$. We first lower bound the likelihood by discarding sequences outside of the set $A(U_h, a)$, so

$$\pi_\theta(a | S_h) \geq$$

$$\sum_{\sigma \in \mathcal{S}_M} \pi_\theta(\text{stop}|a, S_h) \prod_{k=1}^M \pi_\theta(e_k^\sigma | X_{k-1}^\sigma, S_h),$$

where $X_{k-1}^\sigma = [e_1^\sigma ... e_{k-1}^\sigma](U_h)$. This is the same type of sum as in (Shen et al., 2020). Following their derivation,

$$\log \pi_\theta(a | S_h) \geq$$

$$\log \sum_{\sigma \in \mathcal{S}_M} \pi_\theta(\text{stop}|a, S_h) \prod_{k=1}^M \pi_\theta(e_k^\sigma | X_{k-1}^\sigma, S_h) \geq$$

$$C + \frac{1}{M!} \sum_{\sigma \in \mathcal{S}_M} \sum_{k=1}^M \log \pi_\theta(e_k^\sigma | X_{k-1}^\sigma, S_h),$$

where $C = \log(M!) + \log \pi_\theta(\text{stop}|a, S_h)$. The last line comes from Jensen's inequality, where the sum over permutations became an expectation. Using the same trick of rearranging the sum over time steps and orderings as in Shen et al. (2020), we can rewrite the second term as,

$$\frac{1}{M!} \sum_{\sigma \in \mathcal{S}_M} \sum_{k=1}^M \log \pi_\theta(e_k^\sigma | X_{k-1}^\sigma, S_h) =$$

$$\mathbb{E}_k \mathbb{E}_{\sigma_{0:k-1}} \left[ \frac{M}{M-k+1} \sum_{e_k^\sigma} \log \pi_\theta(e_k^\sigma | X_{k-1}^\sigma, S_h) \right]$$

where the sum is over the set $\{e : e_k^\sigma = e, \text{ some } \sigma \in \mathcal{S}_M\}$. We then fold the $\log \pi_\theta(\text{stop}|a, S_h)$ term into the expectation over $k$. This gives the objective,

$$\mathbb{E}_k \mathbb{E}_{\sigma_{0:k-1}} \left[ \frac{M+1}{n_k} \sum_{\sigma_k} \log \pi_\theta(e_k^\sigma | X_{k-1}^\sigma, S_h) \right],$$

where the expectation is now over $k = 1, ..., M+1$, $e_{M+1}^\sigma = \text{stop}$ for all $\sigma \in \mathcal{S}_M$, and

$$n_k = \begin{cases} M - k + 1, & \text{if } k \leq M \\ 1, & \text{if } k = M + 1 \end{cases}$$

**Algorithm 3:** Noisy Edit Trajectory Sampling

**Data:** $a, u_h, \sigma$
**Result:** Edits $e_1, ..., e_M$
$x \leftarrow u_h$;
$k \leftarrow 1$;
**repeat**
    Compute $A(x, a)$;
    Sample $e$ from $A(x, a)_1$;
    $p \leftarrow \text{Bernoulli}(\sigma)$;
    **if** $p = 1$ **then**
        | $e_k \leftarrow$ random edit;
    **else**
        | $e_k \leftarrow e$;
    **end**
    $x \leftarrow [e_k](x)$;
    $k \leftarrow k + 1$;
**until** $x = a$;
**return** $e_1, ..., e_k$

**Algorithm 4:** Token Edit Model Sampling

**Data:** $s, u, \alpha, N$
**Result:** Generation fro $\pi_\theta(\cdot|s)$
$x_0 \leftarrow u$;
$s_0 \leftarrow 0$;
$i \leftarrow 0$;
**repeat**
    $s_i \leftarrow \pi_\theta(\text{stop}|x_i, s)$;
    Sample non-stopping edit $e_i$
      from $\tilde{\pi}_\theta(\cdot|x_i, s)$;
    $x_{i+1} \leftarrow [e_i](x_i)$;
    $i \leftarrow i + 1$;
**until** $s_i > \alpha$ *or* $i > N$;
$j \leftarrow \text{argmax}\{s_i\}$;
**return** $x_j$

## A.6 Denoising

In practice, the lower bound derived above may be very loose because we are only considering edits in $A(U_h, a,)$. In other words, the learned policy $\pi_\theta$ will inevitably find itself out of distribution at inference time by making mistakes.

In order to make the model robust, we leverage denoising training (Lee et al., 2018). To do so, we notice that in Algorithm 2 we are essentially sampling a trajectory of edits $e_1, ...e_M$ such that $[e_1...e_M](U_h) = a$, and then randomly sampling a point along this trajectory. Instead, we sample noisy trajectories of edits, where we intersperse random edits, as in Algorithm 3. This simulates mistakes that the trained policy might make at inference time. Given this random prefix of edits $e_1, ..., e_{k-1}$, we get a noised, intermediate document, $\tilde{x}_{k-1}$. We compute the same loss as before over the set of edits $\{e : \tau_1 = e, \tau \in A(\tilde{x}_{k-1}, a)\}$. This is the set of first elements of edit trajectories from $\tilde{x}_{k-1}$ to $a$, and we'll denote it by $A(\tilde{x}_{k-1}, a)_1$. The objective becomes,

$$\frac{M + 1}{n_k} \sum_{e \in A(\tilde{x}_{k-1}, a)_1} \log \pi_\theta(e|\tilde{x}_{k-1}, S_h),$$

where now $n_k = |A(\tilde{x}_{k-1}, a)_1|$. Intuitively, this is like using a noisy roll-in policy to get to $\tilde{x}_{k-1}$, and then matching an expert policy that can produce $a$.

## A.7 Decoding

To decode from the token editing model we use ancestral sampling with a few modifications. First, when sampling an edit (or stopping action) from $\pi_\theta$, we only sample from the top $k$ edits. Just as for autoregressive models, we found that this improves generation quality, since the low-probability edits are usually of poor quality and will figuratively speaking throw a wrench in the decoding process. For long sequences of edits, the probability of sampling a bad edit also becomes non-negligible.

The model also has a tendency to stop early. Again, this is because the probability of stopping erroneously increases as we sample more and more edits. In contrast to other types of mistakes, stopping early is especially bad because there is no way to recover, as opposed to other mistakes that can be fixed with another edit later on. Therefore, we explicitly avoid sampling the stopping action, and instead decode edits until either the model's stopping probability exceeds a threshold, or we reach a maximum number of edits. We then return the document that had the highest stopping probability. The whole decoding procedure is given in Algorithm 4.

## B Experiment Details

### B.1 Data

We use version 3.0.0 of the CNN/DailyMail dataset from the HuggingFace dataset hub (Lhoest et al., 2021). We take the article summaries, which we split into sentences and then filter to sentences with less than 64 tokens, where the tokens are determined by the BART tokenizer from HuggingFace

(Wolf et al., 2020). We then split the data into train, test, and validation splits, with (approximately) 1M, 55K, and 45K instances respectively.

## B.2 User Simulator Heuristics

The environment of our contextual MDP is determined by the behavior of the user simulator. We consider the following methods for generating user edits:

**Ranking** Given a set of edits, we rank them based on their informativeness. As a measure of informativeness, we use IDF scores.

**Adjacent Edits** If the user simulator simply returns the most informative edits, it will have a tendency to make edits in disparate parts of the text. For example, it might return keywords from the end of the document, when the current draft only covers the start. In a realistic setting, users make edits related to the current draft, they do not preempt the end of the document. Thus, we limit the user simulator to producing adjacent edits, where an edit is adjacent if it is adjacent to a match in the alignment to the target.

**Contiguous Edits** While adjacency will keep the user simulator edits relevant to the current draft, it may still have a tendency to produce a disconnected set of edits. Instead, we limit it to only produce contiguous edits.

**Complete Words** Finally, since the user simulator operates on tokenized text, and the tokens may break up words, we also constrain its edits to complete words.

## B.3 Models and Training

**Implementation** Models were implemented with Transformer architectures with additional MLP layers on top. Specifically, we used the BART base and BART large checkpoints made available through the HuggingFace transformers library (Wolf et al., 2020). The models have (approximately) 140M and 400M parameters respectively.

For fixed document $x$ and state $s$, the token editing model parametrizes a distribution over edits along with a stopping action. We refer to the union as edit actions, which are four tuples $(s, l, o, w)$, where $s \in \{0, 1\}$ indicates the stopping action, and $l, o, w$, are as defined for edits. The probability of an edit action $e = (s, l, o, w)$ is parameterized as a product of four probabilities, depending on three

cases. If $s = 1$, then

$$\pi_\theta(e|x, s) = \pi_\theta^s(1|x, s).$$

If $s = 0, o = \text{del}$, then

$$\pi_\theta(e|x, s) = \pi_\theta^s(0|x, s)\pi_\theta^l(l|x, s)\pi_\theta^o(o|l, x, s).$$

Otherwise,

$$\pi_\theta(e|x, s) =$$
$$\pi_\theta^s(0|x, s)\pi_\theta^l(l|x, s)\pi_\theta^o(o|l, x, s)\pi_\theta^w(w|o, l, x, s).$$

Each of $\pi_\theta^s, \pi_\theta^l, \pi_\theta^o, \pi_\theta^w$ is implemented as a separate MLP head.

**Model Inputs** The history of drafts $S_h$ could become prohibitively large, so we keep track of only the last three drafts, denoted by $U_{h-1}, A_{h-1}$ and $U_h$. This allows the agent to reason about the edits it made at the last step and how the user responded. Additionally, we track all words inserted by the user along the entire history. This is easy to do by marking the relevant words, and allows the agent to know which words in the current draft came from the user. In practice, all these features are represented as a diff. See Figure 2 for an illustration, where the different coloring and markings represent which words were inserted or deleted by the agent or user. Practically, we implement this by adding labels to the tokens in the draft $U_h$. We can easily track these labels for the user and token edit models because they operate by making edits. So if the model or user deletes a word, we can mark it as deleted. For the Seq2Seq model this isn't possible because it returns an entire new draft. For example, we couldn't tell if a word was deleted or substituted for another. Instead, we compute the alignment between $U_{h-1}$ and $A_{h-1}$, from which we can read off which tokens were inserted, deleted or substituted. Because words inserted by the user should likely never be deleted by the agent, we keep the labels on all words inserted by the user so that they persist throughout the entire interaction.

**Training** Models were trained on a single 16GB V100 GPU for 600 iterations with a sampling batch size of $B = 10^4$ and a sampling annealing rate of $\lambda = 0.9$. We also used 300 warmup iterations (where $\lambda$ was not annealed). For the token editing model we used a noise level of $\sigma = 0.3$.

**Decoding** We decode from the S2S autoregressive model using beam search with a beam size of 10. For the token editing model we used top 10 sampling of actions, a stopping probability threshold of $\alpha = 0.95$ and a maximum number of edits of $N = 64$.

### B.4 Metrics

We evaluated generation using BLEU (Papineni et al., 2002) and CHRF (Popović, 2015) with the SacreBLEU implementation (Post, 2018). We also evaluate using BLEURT (Sellam et al., 2020) and BARTSCORE (Yuan et al., 2021), which are model-based metrics that have been shown to correlate well with human judgment on various text generation tasks. As BARTSCORE returns a score interpretable as a log-probability, we report the natural exponent of that score. We also use ROUGE (Lin, 2004) for evaluation as an alternative to BLEU, as its scores tend to be less sensitive to length. In the case of both BLEU and ROUGE, we perform evaluation with their unigram versions (BLEU-1 and ROUGE-1) as they provide interpretability of the results at the token level.