# OpenReview forum: "Interactive Text Generation"
_EMNLP/2023/Conference — EMNLP 2023 Main_

### Official Review · Reviewer_sfhR · 2023-08-02

**Soundness:** 4

**Excitement:**

4: Strong: This paper deepens the understanding of some phenomenon or lowers the barriers to an existing research direction.

**Paper Topic And Main Contributions:**

This paper solves a new end-to-end generation task - interactive text generation. The main contributions are that the proposed model can be trained by interacting with user simulators.

**Questions For The Authors:**

A. Does the current document limit the number of edits when aligned with the target document? If so, it is usually set to a few times. How is this set?
B. How to determine what new words to insert during editing？
C. Whether the problem of user input errors in the interaction process has been considered, and whether the model can fix the errors.


**Reasons To Accept:**

1. This paper proposes a new interactive text generation task and benchmark, constructs the interaction process between agent and simulated user, and designs a new idea for the interactive text generation task.
2. The proposed method utilizes the idea of simulation learning to simulate the user editing process
3. This paper shows the performance of text generated after each generation of interaction, and clearly shows that the effect of text generated gradually becomes better with the increase of the number of interactions

**Reasons To Reject:**

1. The description of the model is not clear enough, and there are some confusing points.
2. It does not take into account the error of real users in the input process and lacks the robustness study of the model.
3. For the part of the imitation learning model, the details such as ensuring accuracy are not clearly presented.


**Reproducibility:**

4: Could mostly reproduce the results, but there may be some variation because of sample variance or minor variations in their interpretation of the protocol or method.

**Reviewer Confidence:**

3: Pretty sure, but there's a chance I missed something. Although I have a good feel for this area in general, I did not carefully check the paper's details, e.g., the math, experimental design, or novelty.

---

> ### Author Rebuttal · Authors · 2023-08-29
>
> Thank you for your detailed review of our work. To address your questions and comments:
>
> 1. “The description of the model is not clear enough [...]”
>
> We acknowledge that many technical points needed to be relegated to the appendix in the interest of saving space, however we will use the additional space provided for revisions to clarify any ambiguous points in the main paper.
>
> 2. “It does not take into account the error of real users in the input process and lacks the robustness study of the model.”
>
> We discuss the point of user error below in response to your other question. Concerning robustness, we do include ablations in Appendix C, which may address your concern.  We will also move those results back into the main paper using the additional space provided for revisions.
>
> 3. “For the part of the imitation learning model, the details such as ensuring accuracy are not clearly presented.”
>
> As mentioned in our reply to your first point, we will move some of the discussion of more technical points from the appendix back into the main paper to make the presentation clearer and easier to follow.
>
> 4. “Does the current document limit the number of edits when aligned with the target document? [...]”
>
> The limit on the number of edits was only set for the purpose of a fair comparison between interactive and non-interactive settings. It was set based on the average length of the sentences in the dataset, and for the simple practical reason that N=6 is divisible by both 3 and 2. In a real user scenario, there would be no such limit and the user would be free to make as little or as many edits as they choose.
>
> 5. “How to determine what new words to insert during editing?”
>
> If you are referring to the training of the token editing model, this is inferred using an alignment of the current draft to the goal document, from which we can derive sequences of edits that transform the draft into the goal. This is explained in more detail in Appendix A.3.
>
> 6. “Whether the problem of user input errors in the interaction process has been considered, and whether the model can fix the errors.”
>
> This is an interesting point about user errors. The user simulators we use do not make mistakes, so the model learns to trust the edits made by the user and therefore rarely corrects the user. However, it would in principle be possible to simulate errors from the user, much in the same way that we simulate errors from the model so that it learns to correct itself (Appendix A.6).
>
> Thank you again for your review, and we hope the answers and proposed changes above address your questions and concerns.

---

### Official Review · Reviewer_HjwP · 2023-08-02

**Typos Grammar Style And Presentation Improvements:** 1. I found it confusing to say 'seque…
**Soundness:** 4

**Excitement:**

3: Ambivalent: It has merits (e.g., it reports state-of-the-art results, the idea is nice), but there are key weaknesses (e.g., it describes incremental work), and it can significantly benefit from another round of revision. However, I won't object to accepting it if my co-reviewers champion it.

**Missing References:**

While the work does discuss some interactive NLP work, there are a number of collaborative writing works which also fit the pairwise writing setup [1,2,3] and could be discussed in the context of the motivation for the work. These vary in the style of interaction but as this work aims for a broad task definition of interactive generation, it needs more discussion in this line. In addition, recent work in HCI has also made strides in discussing how we evaluate interactive generation tasks [4,5].

[1] Akoury, Nader, et al. "Storium: A dataset and evaluation platform for machine-in-the-loop story generation." arXiv preprint arXiv:2010.01717 (2020).
[2] Clark, Elizabeth, et al. "Creative writing with a machine in the loop: Case studies on slogans and stories." 23rd International Conference on Intelligent User Interfaces. 2018.
[3] Du, Wanyu, et al. "Read, revise, repeat: A system demonstration for human-in-the-loop iterative text revision." arXiv preprint arXiv:2204.03685 (2022).
[4] Lee, Mina, et al. "Evaluating human-language model interaction." arXiv preprint arXiv:2212.09746 (2022).
[5] Shen, Hua, and Tongshuang Wu. "Parachute: Evaluating interactive human-lm co-writing systems." arXiv preprint arXiv:2303.06333 (2023).

**Paper Topic And Main Contributions:**

This paper makes the case that we often do not train/optimize user-centric machine learning models to perform well interactively. The authors' claim is that what we really want in this setting is a model that adapts well to user instructions to obtain the desired output (not necessarily one that gets as close as possible in a one-shot manner).

(Contributions)
Hence the authors formulate an interactive text-generation task setting where the model also receives a fixed budget of edits from a 'user' along with the task input/prompt in order to guide the generation process.
In this regime, the authors are able to compare interactive and non-interactive models by varying when the 'user edits' are received - in the non-interactive case, all edits are received at the start.
The authors then present experiments evaluating various models in the interactive vs. non-interactive setting on the task of summarization. In order to do so, they also create a simple user simulator to provide these 'edits' which are essentially a form of oracle information during the generation. The findings show that interactivity helps across model architectures (autoregressive and seq2seq transformer models) and across various user simulators.

**Questions For The Authors:**

1. I'm curious about the choice of N being fixed for a given task because the relationship between the total budget of edits and the length of the document is likely non-linear. For instance, if you are creating a longer document, then you might see a much greater benefit from a larger editing budget than a shorter document (in the interactive case because you can guide the document more as opposed to providing a longer initial draft).

2. Does increasing the number of episodes in Fig. 4 cause the lines to taper off i.e. is there a limit to the benefit of interactivity?

3. The choice of summarization is apt for the interactive generation due to the subjective nature of relevance for the task. However, I think that the chosen datasets do not best fit this need. Perhaps query focused summarization datasets like Squality could be interesting to work with?

**Reasons To Accept:**

Given some assumptions made in the formulation (discussed in Weakness 1), the paper provides a clear formulation for the interactive task setting and an accompanying user simulator/agent training algorithm.

While I did mention some missing references, I think the novelty of providing a means of comparison between interactive and non-interactive generation (given some assumptions) is great.

The training of the editor models (section 4.3) more directly matches the desired interactive evaluation which results in clear performance improvements over the non-interactive baseline.

**Reasons To Reject:**

In order to compare interactive and non-interactive models, the authors make an assumption that all edits are equal regardless of when they are provided in the writing process. This is not always true as some key edits provided once some progress toward the final draft is made could influence a document more than just an initial draft to begin with.


(Minor) There's a mismatch between the motivating examples and the formulation in Section 2. In lines 121-126, the examples given are a student writing with an advisor/freelancer obtaining feedback from their client. In each of these cases, the latter is the 'user'. Then the formulation in lines 137-143 describes that the user knows the oracle document which is the target of the generation process. I think that this is not always realistic. Perhaps this is true only in a student/teacher setting and in each of the editing examples provided, the user is not necessarily an oracle but just an agent that exercises preferences (which are subjective and malleable based on the generated documents and not objective oracle) in the editing process.


(Minor) While generally missing references are not a 'reason to reject', I do think that since this work is positioned as a new task setting, it is important to discuss existing work in the collaborative writing space and the evaluation of interactive systems from the intersection of HCI/NLP.





**Reproducibility:**

3: Could reproduce the results with some difficulty. The settings of parameters are underspecified or subjectively determined; the training/evaluation data are not widely available.

**Reviewer Confidence:**

2: Willing to defend my evaluation, but it is fairly likely that I missed some details, didn't understand some central points, or can't be sure about the novelty of the work.

---

> ### Author Rebuttal · Authors · 2023-08-29
>
> Thank you for your detailed review of our work. To address your questions and comments:
>
> 1. “[...] the authors make an assumption that all edits are equal regardless of when they are provided in the writing process. [...]”
>
> You are correct that the quality of an edit depends on when it is given in the generation process. This is in fact one of the points that we would like to make. In our evaluation, we show that solely by letting a model receive edits from the user intermittently, rather than all at the start, the model is able to produce a better document. This should clearly demonstrate that getting edits later can be beneficial, which justifies the use of interactive generation models over one-shot models.
>
> 2. “[...] it is important to discuss existing work in the collaborative writing space and the evaluation of interactive systems from the intersection of HCI/NLP.”
>
> We agree that it would be beneficial to have a broader discussion of related work and we will add and discuss the additional references you suggest. However, we do wish to note that our goal in this work was to formulate a very concrete framework for interactive generation, so we believe our work to be orthogonal, but complementary, to work at the intersection of HCI/NLP.
>
> 3. “I'm curious about the choice of N being fixed for a given task [...]”
>
> You are correct that N should vary with respect to the task length. In our evaluations, we keep N fixed to ensure fair comparisons between interactive and non-interactive settings. However, the task does not otherwise assume a fixed number of edits (for technical reasons there is a fixed horizon H, but this can be very large). With a real user, the interaction stops whenever the user is satisfied with the document, so it may go on for as long as needed.
>
> 4. “Does increasing the number of episodes in Fig. 4 cause the lines to taper off [...]”
>
> In our evaluation setup we do expect to see a taper. We fixed the total number N of edits made by the user  to ensure a fair comparison. As the number of episodes M increases, the number of edits N/M made by the user in each episode decreases. Eventually, this may be so small that the model cannot gain any useful information from the user anymore. For example, if the user makes a single edit to a particle word (e.g. they change "a" to "an"), this would not convey much information.
>
> 5. “[...] Perhaps query focused summarization datasets like Squality could be interesting to work with?”
>
> A summarization dataset such as Squality could be interesting to work with, but it would require an additional component of conditioning on the text to be summarized, which we believe falls out of scope for this initial work. However, this could make for an exciting future direction.
>
> 6. “I found it confusing to say 'sequence to sequence autoregressive model' (l.263, 405) [...]”
>
> We will change the terminology from sequence to sequence autoregressive model to left-to-right autoregressive model which should be more apt.
>
> 7. “In l.165, what is the 'resp user.' meant to convey?”
>
> We will rewrite line 165 to avoid the use of resp. (short for respectively).
>
> Thank you again for your review, and we hope the answers and proposed changes above address your questions and concerns.

---

### Official Review · Reviewer_tSro · 2023-08-11

**Soundness:** 4

**Excitement:**

4: Strong: This paper deepens the understanding of some phenomenon or lowers the barriers to an existing research direction.

**Justification For Ethical Concerns:**

-

**Missing References:**

Recent works exploring multi-turn interactivity using LLMs
@inproceedings{
    sun2023recitationaugmented,
    title={Recitation-Augmented Language Models},
    author={Zhiqing Sun and Xuezhi Wang and Yi Tay and Yiming Yang and Denny Zhou},
    booktitle={The Eleventh International Conference on Learning Representations },
    year={2023},
    url={https://openreview.net/forum?id=-cqvvvb-NkI}
}

@misc{bang2023multitask,
      title={A Multitask, Multilingual, Multimodal Evaluation of ChatGPT on Reasoning, Hallucination, and Interactivity},
      author={Yejin Bang and Samuel Cahyawijaya and Nayeon Lee and Wenliang Dai and Dan Su and Bryan Wilie and Holy Lovenia and Ziwei Ji and Tiezheng Yu and Willy Chung and Quyet V. Do and Yan Xu and Pascale Fung},
      year={2023},
      eprint={2302.04023},
      archivePrefix={arXiv},
      primaryClass={cs.CL}
}

@misc{huang2022large,
      title={Large Language Models Can Self-Improve},
      author={Jiaxin Huang and Shixiang Shane Gu and Le Hou and Yuexin Wu and Xuezhi Wang and Hongkun Yu and Jiawei Han},
      year={2022},
      eprint={2210.11610},
      archivePrefix={arXiv},
      primaryClass={cs.CL}
}

@misc{gou2023critic,
      title={CRITIC: Large Language Models Can Self-Correct with Tool-Interactive Critiquing},
      author={Zhibin Gou and Zhihong Shao and Yeyun Gong and Yelong Shen and Yujiu Yang and Nan Duan and Weizhu Chen},
      year={2023},
      eprint={2305.11738},
      archivePrefix={arXiv},
      primaryClass={cs.CL}
}

**Paper Topic And Main Contributions:**

The paper introduces a new task and framework that requires user-system interactivity namely interactive text generation. The task and framework requires the model to generates a document similar to the target document G, starting from a blank document A0. Instead of using a real user for interactivity, the work incorporate a user simulator (oracle model) which produces a revision list Uh given the generated draft document Ahand the gold document G. The paper showcases two imitation learning models to iteratively generate draft, i.e., Left to Right Autoregressive Model and Token Editing Model. The autoregressive model directly generates a new draft Ah using the draft and revision history Sh in an autoregressive manner, while the token editing model perform iterative word-to-word editing until it achieves a certain stopping criterion. The effectiveness of the framework is tested on single sentence generation retrieved from the article summaries of the CNN/DailyMail dataset. The results suggest that: 1) token editing model can be comparable to the seq2seq baselines while being more naturally suited to the task, 2) the interactive system shows better semantic similarity with the gold, while having a slightly lower fluency score.

**Questions For The Authors:**

- Given the expert policy always produces the goal G, what is the different between using the expert policy and simply using supervise objective to train the model for each turn?
- How does the expert policy implemented for the token editing model?
- For the seq2seq baseline, what is the size of the model? Given that there are two model variants for token editing model, i.e., editor and editor large, readers would expect the two variants are also incorporated for the baseline to get a better comparison.

**Reasons To Accept:**

- Although many papers have showcased that interactivity can improve generation results, this paper introduces a formalized framework to train and evaluate model interactivity
- The paper introduces a token editing model that generates revised documents via iterative token editing steps. The paper also provides a thorough evaluation compared to the seq2seq baseline

**Reasons To Reject:**

- Although the paper mentions the scoring function s(·; G) can be generalized for multiple goal documents or even without any gold document, such generalization is not explored which makes the generalization of the framework itself questionable.
- Many important details of the paper are not incorporated in the main paper, which makes it difficult to follow the paper without thoroughly reading the appendix.
- The paper claims that the work contributes different user simulators and the performance of the model remains consistent with different user simulators. This contribution is nowhere to be found in the main paper.
- The experiment is only conducted on smaller-scale LM which is not instruction-tuned, despite many prior works working on a similar problem exploring interactivity using much larger LMs.

**Reproducibility:**

3: Could reproduce the results with some difficulty. The settings of parameters are underspecified or subjectively determined; the training/evaluation data are not widely available.

**Reviewer Confidence:**

4: Quite sure. I tried to check the important points carefully. It's unlikely, though conceivable, that I missed something that should affect my ratings.

**Typos Grammar Style And Presentation Improvements:**

- Line 164--165: If we denote the agent, resp. user, draft at step h by Ah, resp. Uh ... => incomprehensible sentence & the term "resp." is unclear
- In the oracle model of Figure 2, it will be easier to understand if it is stated which one is Ah and which one is Uh
- Appendix C is never refered in the main paper.

---

> ### Author Rebuttal · Authors · 2023-08-29
>
> Thank you for your detailed review of our work. To address your questions and comments:
>
> 1. “Many important details of the paper are not incorporated in the main paper [...]”
>
> We acknowledge that many technical points were left to the appendix. Because our paper is introducing a new framework for interactive generation, we opted to use most of the available space to discuss the framework, while moving more technical details to the appendix. Given the additional space provided for revisions, we will clarify these technical points directly in the main paper.
>
> 2. “The paper claims that the work contributes different user simulators [...]”
>
> The different user simulators come from the different user heuristics described in Appendix B.2. These can be combined or tuned in various ways to produce different user simulator behaviors. We evaluate with different user simulators in Appendix C, Table 4. However, as you note, this appendix is not referred to in the text. We will move these results back into the main paper, or make more explicit reference to them and clarify our claim.
>
> 3. “Given the expert policy always produces the goal G, what is the different between using the expert policy and simply using supervise objective [...]”
>
> You are correct that because the expert is quite simple, this is equivalent to training with a supervised objective. However, the main reason to use the expert is for the roll-in policy, which is a mixture between the expert and the current trained policy.
>
> 4. “How does the expert policy implemented for the token editing model?”
>
> The expert policy, as referred to in our answer in point 3, is the same for all models. However, the way that the token editing model is trained to produce drafts can itself be seen as a form of imitation learning, although we do not explicitly frame it that way in our paper. If you are referring to the expert in this context, it uses an alignment between the current draft and the goal document to determine a set of edit trajectories from the draft to the goal. This is akin to an expert policy in a driving task that dynamically finds a route from the vehicle's current position to a target position. This is explained in full detail in appendix A.3.
>
> 5. “For the seq2seq baseline, what is the size of the model? [...]”
>
> The seq2seq baseline (S2S) and the token editing model (Editor) are the same size. Both are based on the BART base checkpoint from Huggingface. Technically, the seq2seq baseline is actually larger since it uses both the encoder and decoder, while the token editing model only uses the encoder. Full implementation details are given in Appendix B.3, but we will clarify this point in the main text.
>
> 6. Missing references
>
> We will include the additional references you mentioned in the related work. We do however wish to note that, while these other works exploring interactivity are certainly important to discuss, we believe our work explores an orthogonal direction that has not yet been considered, since we formalize a concrete framework for interactivity that is used not just at inference time but also at training time.
>
> 7. “Line 164--165: [...] unclear”
>
> We will rewrite the sentence in lines 164-165 to make it clearer
>
> 8. “In the oracle model of Figure 2, it will be easier to understand if it is stated which one is Ah and which one is Uh”
>
> We will clarify in Figure 2, in the Oracle Model box, which document is A_h and which is U_{h+1}
>
> 9. “Appendix C is never refered in the main paper.”
>
> As noted in point 2, we will move appendix C back into the main paper, or make the reference more explicit.
>
> Thank you again for your review, and we hope the answers and proposed changes above address your questions and concerns.

---

### Meta-Review · Area_Chair_QE8G · 2023-09-18

**Recommendation:** 5

**Metareview:**

This paper proposes a novel framework for interactive text generation and introduces a new task benchmark. The reviewers agree that this framework and benchmark are useful for interactive text generation. The paper also proposes an interactive text-editing model and provides a thorough analysis of the proposed method. One area for improvement raised by the reviewers is that some of the important technical details are buried in the appendix. It would be beneficial to include these in the main body of the paper, a suggestion with which the authors have agreed, given the extra page.

---

### Decision · Program_Chairs · 2023-10-07

**Decision:**

Accept-Main

**Comment:**

This paper proposes a novel framework for interactive text generation and introduces a new task benchmark. The reviewers agree that this framework and benchmark are useful for interactive text generation. The paper also proposes an interactive text-editing model and provides a thorough analysis of the proposed method. One area for improvement raised by the reviewers is that some of the important technical details are buried in the appendix. It would be beneficial to include these in the main body of the paper, a suggestion with which the authors have agreed, given the extra page.